# Technical Report for ICML 2024 Automated Math Reasoning Challenge: Solving Optimization Problems with Open Source Large Language Model

**Duc M. Nguyen** [1]  **Sungahn Ko** [1]

## Abstract

This technical report presents an approach utilizing open-source Large Language Models for Automated Optimization Problem-solving With Code Challenge at the ICML 2024 AI4Math Workshop. This challenge emphasizes the ability of Large Language Models (LLMs) to handle complex mathematical reasoning from formulating to solving the problem at hand. By exploring different prompting techniques, such as few-shot, self-consistency, chain-of-thought, and tree-of-thought, we aim to explore the current state-of-the-art LLMs' mathematical reasoning abilities.

## 1. Introduction

Recent research has highlighted the remarkable potential of state-of-the-art Large Language Models like GPT-4 (Achiam et al., 2023) showcasing their promising abilities in reasoning across diverse fields, encompassing tasks such as solving mathematical word problems and proving theorems (Huang et al., 2024). Automated mathematical reasoning, which requires sophisticated multi-step planning and reasoning, has attracted active research to evaluate and develop intelligent agents capable of obtaining advanced forms of human intelligence such as mathematical reasoning.

In this technical report, we investigate the ability to formulate and solve optimization problems, which is critical across various domains, ranging from operations research and engineering to finance and machine learning, by Open Source Large Language Models. Traditionally, solving optimization problems has required human expertise in mathematical modeling and algorithm design. However, the rise of LLMs presents an opportunity to automate this process, enabling machines to understand, interpret, and solve optimization problems expressed in natural language. To tackle this problem, we investigate current Large Language Models' capability to solve optimization problems by augmenting them with various prompt engineering frameworks. Our contributions can be summarized as follows:

- We formulate the automated optimization problem-solving as a prompt engineering problem, leveraging the capabilities of LLMs to understand and model optimization problems expressed in natural language.

- We propose a symbolic approach to repair incorrect problem formulation automatically, improving the accuracy and reliability of the models.

- We provide a comparative analysis of state-of-the-art prompting techniques such as few-shot, self-consistency, chain-of-thought, and tree-of-thought, offering insights into their strengths and limitations in automated optimization problem-solving.

## 2. Related Work

### 2.1. Large Language Models

In the past few years, the boom of deep learning in Natural Language Processing and the birth of the Transformers (Vaswani et al., 2017) architecture dictates the trend in how language models are designed. Bigger and deeper models are preferred since they are capable of producing human-like text and achieving state-of-the-art performance on most of the benchmarks. In 2023, Achiam et al. (2023) transformed the world of NLP research with the release of GPT-4 with unprecedented state-of-the-art overperformance on most benchmarks. Following their success, Anthropic AI released a competing series of models, namely Claude 1, 2, and 3, with similar performance (Anthropic, 2023). However, they were closed-source and people had to pay for API access.

To provide an open-source alternative, numerous researchers have attempted to train their own LLM and published their findings. One of the earliest open-source LLM, LLaMA (Touvron et al., 2023a), was released by Meta. It inspired others to develop better models based on the LLaMA architecture. BigScience Workshop (2022) introduced BLOOM,

[1]Department of Computer Science and Engineering, Ulsan National Institute of Science and Technology, Ulsan, Republic of Korea. Correspondence to: Sungahn Ko <sako@unist.ac.kr>.

*The first AI for Math Workshop at the $41^{st}$ International Conference on Machine Learning*, Vienna, Austria. Copyright 2024 by the authors.

an open-source LLM with similar sizes and performance to GPT-3. Shliazhko et al. (2022) reproduced the GPT-3 architecture using GPT-2 sources and introduced mGPT with multilingual capability. Touvron et al. (2023b) improved the previous LLaMA by introducing LLaMA 2 with a focus on instruction fine-tuning. The model approached GPT-3's performance but ten times smaller and faster. Deepseek AI introduced DeepSeek-LLM (Bi et al., 2024), achieving superior performance compared to GPT-3.5, and its programming variant DeepSeek-Coder (Guo et al., 2024), achieving the state-of-the-art performance in programming and logical reasoning among both open and closed source models apart from GPT-4. Recently, Meta released LLaMA 3 (lla), the best overall performing model among open-source alternatives.

### 2.2. Automated Reasoning through Prompt Engineering

Automated reasoning in the context of large language models has gained significant attention due to the advancements in prompt engineering techniques. These techniques allow LLMs to perform complex reasoning tasks by structuring prompts in a way that guides the model to generate desired outputs. The primary goal of prompt engineering is to leverage the pre-trained knowledge of LLMs and coax them into producing accurate and coherent responses without the need for additional fine-tuning.

Early work in prompt engineering focused on simple task-specific prompts. Brown et al. (2020) introduced few-shot learning with GPT-3, where the model is given a few examples of the task at hand within the prompt, demonstrating its capability to generalize from minimal context. This technique was extended to zero-shot and one-shot learning scenarios, showing that LLMs could perform well on tasks with little to no example-specific data.

Subsequent research explored more sophisticated prompt engineering methods. Wei et al. (2022a) showed that letting the LLM produce intermediate steps (chain of thoughts) significantly boosts the model's performance on non-trivial output tasks such as mathematical problem solving, logical reasoning, and commonsense question answering. This chain-of-thought (COT) prompting method guides the model through a sequence of intermediate reasoning steps, effectively breaking down complex problems into more manageable parts. This approach not only improves the accuracy of the responses but also provides a transparent view of the model's reasoning process. Wang et al. (2022) introduced the self-consistency decoding strategy, greatly improving COT performance by generating multiple reasoning paths and selecting the most consistent one using majority voting. Recently, Yao et al. (2023) proposed a novel framework for language model inference, Tree of Thoughts, which expands on COT by exploring multiple reasoning paths in a

tree-like structure, further enhancing the model's problem-solving abilities by considering various possible solutions and converging on the most optimal one.

## 3. Approaches

### 3.1. Problem Formulation

In this section, we provide a formal formulation of Automated Optimization Problem Solving.

**Definition 3.1.** *Input Sequence.* Let $P = (p_1, p_2, \ldots, p_n)$ be an ordered set of tokens that represents the optimization problem statement, where $n$ is the length of the problem, $p_i \in \mathbb{V}$ and $\mathbb{V}$ is the set of vocabulary.

**Definition 3.2.** *Output Sequence.* Let $C = (c_1, c_2, \ldots, c_m)$ be an ordered set of tokens that represents the solution to the optimization problem, where $m$ is the length of the solution, $c_i \in \mathbb{V}$ and $\mathbb{V}$ is the set of vocabulary.

**Definition 3.3.** *Objective Function.* Given the problem statement $P = (p_1, p_2, \ldots, p_n)$, let $\mathcal{F} : \mathbb{V}^{n \times 1} \to \mathbb{V}^{m \times 1}$ be a function that produces a solution to $P$, i.e., $\mathcal{F}(P) = C$ should be the correct solution to $P$. Our objective is to find (approximate) the function $\mathcal{F}$.

**Definition 3.4.** *Prompt Format.* Let $S = (s_1, s_2, \ldots, s_{n_s})$, $U = (u_1, u_2, \ldots, u_{n_u})$, and $A = (a_1, a_2, \ldots, a_{n_a})$ be ordered sets representing sequences of the system text, user text, and assistant text where $n_s, n_u, n_a$ are the length of the system, user, assistant respectively, and $s_i, u_i, a_i \in \mathbb{V}$. The historical conversation is a list of these sequences, denoted as $\mathcal{P} = [S, U^{(1)}, A^{(1)}, U^{(2)}, A^{(2)}, U^{(3)}, \ldots]$, where $\mathcal{P}_1$ is the system prompt and $\mathcal{P}_i$ will alternate between $U$ and $A$ for $i > 1$.

**Definition 3.5.** *Reasoning Path.* Let $R = (r_1, r_2, \ldots, r_{n_r})$ be an ordered set representing the sequence of intermediate reasoning steps, where $n_r$ is the length of the reasoning path and $r_i \in \mathbb{V}$.

**Definition 3.6.** *Reasoning Path Evaluation.* Let's define a function $\mathcal{E} : \mathbb{V}^{n_r \times 1} \to \mathbb{R}$ to be an evaluator of a reasoning path $R$ that gives a score representing how good the reasoning is.

Code data have been demonstrated to significantly enhance the reasoning capabilities of large language models (LLMs) by providing structured and logical examples that improve their problem-solving skills (Ma et al., 2024). Given this advantage, we restrict the output sequence in Definition 3.2 to be Python source code in this technical report. This restriction allows us to leverage the benefits of code data, ensuring that the generated solutions are executable and verifiable. By focusing on Python, we can take advantage of its simplicity and the extensive range of libraries available for numerical optimization and algorithm development, facilitating efficient implementation and testing of our automated

reasoning framework. Examples of the prompt template can be found in Appendix A.

## 3.2. Automated Optimization Problem Solving

In this section, we discuss the main approaches to tackle this problem.

**Few-shot Prompting**  Few-shot prompting involves providing the model with a small number of examples within the prompt to guide it toward the correct solution. Formally, let $\mathcal{D}_{\text{train}} = \{(P_i, C_i)\}_{i=1}^{k}$ be a set of $k$ training examples, where each $P_i$ is an input sequence representing an optimization problem statement, and $C_i$ is the corresponding output sequence representing the solution.

Given a new optimization problem $P_{\text{new}}$ and a System instruction, we construct the prompt $\mathcal{P}$ based on Definition 3.4 as follows:

$$\mathcal{P} = [S, P_1, C_1, P_2, C_2, \ldots, P_k, C_k, P_{\text{new}}] \quad (1)$$

The model $\mathcal{M}$ then generates the solution $\hat{C}_{\text{new}}$ for the new problem $P_{\text{new}}$ based on the constructed prompt $\mathcal{P}$:

$$\hat{C}_{\text{new}} = \mathcal{M}(\mathcal{P})$$

Incorporating few-shot prompting allows the model to leverage examples to infer patterns and structures that are useful for solving the new problem, effectively reducing the need for extensive fine-tuning and enabling the model to generalize from minimal context.

**Self-consistency Prompting**  Self-consistency prompting builds on the concept of few-shot prompting by introducing multiple reasoning paths and selecting the most consistent one to enhance the reliability of the model's outputs. Formally, given the prompt $\mathcal{P}$ constructed as in Equation 1, the model generates multiple candidate solutions $\{\hat{C}_{\text{new}}^{(j)}\}_{j=1}^{M}$, where $M$ is the number of generated solutions.

Each candidate solution $\hat{C}_{\text{new}}^{(j)}$ is evaluated for consistency. The final solution $\hat{C}_{\text{new}}^{\text{final}}$ is selected based on a majority voting mechanism:

$$\hat{C}_{\text{new}}^{\text{final}} = \text{mode}\left(\{\hat{C}_{\text{new}}^{(j)}\}_{j=1}^{M}\right)$$

This approach improves the robustness of the generated solutions by considering multiple possibilities and converging on the most consistent answer, thereby enhancing the accuracy and reliability of the model in solving optimization problems.

**Chain-of-thought Prompting**  Chain-of-thought (CoT) prompting involves guiding the model through a sequence of intermediate reasoning steps, enabling it to break down complex problems into more manageable parts. This approach enhances the model's problem-solving capabilities by making the reasoning process explicit and structured.

Given an optimization problem $P$, we augment the assistant text $A$ to include intermediate steps. Let $A_{\text{CoT}}$ be the new assistant text that includes the intermediate reasoning steps $R$, and the final solution source code $C$. Formally, $A_{\text{CoT}} = (R, C)$. Then, given a new optimization problem $P_{\text{new}}$, the chain-of-thought prompt $\mathcal{P}_{\text{COT}}$ is

$$\mathcal{P}_{\text{CoT}} = [S, P_1, A_{\text{CoT}_1}, P_2, A_{\text{CoT}_2}, \ldots, P_k, A_{\text{CoT}_k}, P_{\text{new}}] \quad (2)$$

The model $\mathcal{M}$ then generates the intermediate steps $\hat{R}_{\text{new}}$ and the final solution $\hat{C}_{\text{new}}$ as follows:

$$\hat{A}_{\text{CoT}_{\text{new}}} = (\hat{R}_{\text{new}}, \hat{C}_{\text{new}}) = \mathcal{M}(\mathcal{P}_{\text{CoT}})$$

We can combine COT with self-consistency by letting the model $\mathcal{M}$ generate multiple candidates $\{\hat{A}_{\text{CoT}_{\text{new}}}^{(j)}\}_{j=1}^{M}$, then selecting the final solution $\hat{A}_{\text{CoT}_{\text{new}}}^{\text{final}}$ with majority voting

$$\hat{A}_{\text{CoT}_{\text{new}}}^{\text{final}} = \text{mode}\left(\{\hat{A}_{\text{CoT}_{\text{new}}}^{(j)}\}_{j=1}^{M}\right)$$

By including these intermediate reasoning steps, CoT prompting helps the model to decompose the problem into smaller, more manageable parts, thereby improving its ability to generate accurate and coherent solutions. This method not only enhances the model's performance on complex tasks but also provides transparency into the reasoning process, making it easier to understand and verify the generated solutions.

**Tree-of-thought Prompting**  Tree-of-thought (ToT) prompting extends the concept of chain-of-thought (CoT) prompting by exploring multiple reasoning paths in a tree-like structure. This approach allows the model to consider various possible solutions and converge on the most optimal one, further enhancing its problem-solving capabilities. The original ToT algorithm integrates Depth-First-Search (DFS), and Breadth-First-Search (BFS) into the framework to search for the best reasoning path. Given an optimization problem $P_{\text{new}}$, we construct the tree-of-thought prompt $\mathcal{P}_{\text{ToT}}$ similarly to Equation 2 as follows

$$\mathcal{P}_{\text{ToT}} = [S, P_1, A_{\text{CoT}_1}, P_2, A_{\text{CoT}_2}, \ldots, P_k, A_{\text{CoT}_k}, P_{\text{new}}] \quad (3)$$

In this technical report, we investigate the Beam Search strategy to enhance the ToT framework. Beam Search is a heuristic search algorithm that explores a graph by expanding the most promising nodes in a limited set, known as the beam width, at each level. To obtain promising candidates,

we utilize a neural evaluator $\mathcal{E}$ defined in Definition 3.6. The evaluator $\mathcal{E}$ is composed of the same LLM used for generating solutions, and a detailed instruction to instruct the model to grade the candidate solutions. This strategy balances the exploration and exploitation of the search space, maintaining a fixed number of best candidates while pruning less promising paths (Lowerre, 1976). The algorithm is summarized in Algorithm 1.

Let $b_1$ and $b_2$ be the beam width, representing the number of top candidates retained at each level of the tree, and the number of samples to generate for each retained candidate respectively. The Beam Search of Thought (BSOT) is as follows.

1. Start with the root node representing the prompt "*Let's think step by step*" to initiate a chain-of-thought answer. This prompt is a common practice prior works use to initiate the chain-of-thought mechanism (Kojima et al., 2022).

2. The model $\mathcal{M}$ then generates $b_2$ intermediate reasoning steps $\{R_{i,1}, R_{i,2}, \ldots, R_{i,b_2}\}$ using the model $\mathcal{M}$ for each candidate at the current level.

3. For each generated reasoning step, concatenate with the previous path $A'_{\text{ToT}}$ and obtain evaluations from the evaluator $\mathcal{E}$.

4. Retain the top $b_1$ candidates among all generated samples.

5. Repeat this process up to a specified depth $d$ or until $b_1 \times b_2$ terminal nodes $\{\hat{A}^{(j)}_{\text{TOT}_{\text{new}}}\}_{j=1}^{b_1 b_2}$ are obtained.

The final solution $\hat{A}^{\text{final}}_{\text{TOT}_{\text{new}}}$ is selected based on a majority voting mechanism:

$$\hat{A}^{\text{final}}_{\text{TOT}_{\text{new}}} = \text{mode}\left(\{\hat{A}^{(j)}_{\text{TOT}_{\text{new}}}\}_{j=1}^{b_1 b_2}\right)$$

By incorporating Beam Search into the ToT framework, we aim to improve the efficiency and effectiveness of the model's reasoning process. This method allows the model to navigate the solution space more intelligently, retaining the most promising paths while discarding less relevant ones.

**Ensemble Strategy**   It has been shown that combining the outputs of multiple models can significantly improve performance and robustness. This approach, known as the ensemble strategy, leverages the strengths of different models to produce more accurate and reliable results (Sagi & Rokach, 2018). Formally, let $\{\hat{\mathcal{F}}_i\}_{i=1}^{N}$ be a set of $N$ diverse estimator of the function $\mathcal{F}$, each trained or configured differently. Given an optimization problem $P_{\text{new}}$, each model

---

**Algorithm 1** The BSOT Algorithm

1: **Input:** Problem statement $P_{\text{new}}$, beam widths $b_1, b_2$, and maximum depth $d$
2: **Output:** Solution $\hat{A}^{\text{final}}_{\text{TOT}_{\text{new}}}$
3: $root := \mathcal{P}_{\text{TOT}} + [\text{"}Let's\ think\ steps\ by\ steps\text{"}]$
4: Initialize the beam $\mathcal{B}_1 = \{root\}$
5: **for** $i = 1$ to $d$ **do**
6:    Initialize $\mathcal{B}_{i+1} = \{\}$
7:    **for** each path $T$ in $\mathcal{B}_i$ **do**
8:       Generate candidate reasoning steps

$$\left\{R_{i,j} \Big| R_{i,j} = \mathcal{M}(T), 1 \le j \le b_2\right\}$$

9:       **for** each candidate $R_{i,j}$ **do**
10:          Append $R_{i,j}$ to path $T$ to form new path $T'$
11:          Add $T'$ to $\mathcal{B}_{i+1}$
12:       **end for**
13:    **end for**
14:    Sort $\mathcal{B}_{i+1}$ by $\mathcal{E}(T')$ and retain top $b_1$ paths
15: **end for**
16: Obtain candidate solutions $\{\hat{A}^{(j)}_{\text{TOT}_{\text{new}}}\}_{j=1}^{|\mathcal{B}_d|}$ from the final beam $\mathcal{B}_d$
17: Select the final solution $\hat{A}^{\text{final}}_{\text{TOT}_{\text{new}}}$ based on majority voting:

$$\hat{A}^{\text{final}}_{\text{TOT}_{\text{new}}} = \text{mode}\left(\{\hat{A}^{(j)}_{\text{TOT}_{\text{new}}}\}_{j=1}^{|\mathcal{B}_d|}\right)$$

18: **return** $\hat{A}^{\text{final}}_{\text{TOT}_{\text{new}}}$

---

$\hat{\mathcal{F}}_i$ generates a candidate solution $\hat{C}^{(i)}_{\text{new}}$:

$$\hat{C}^{(i)}_{\text{new}} = \hat{\mathcal{F}}_i(P_{\text{new}})$$

The ensemble strategy combines these candidate solutions to produce the final solution $\hat{C}^{\text{final}}_{\text{new}}$. One common method is to use a voting mechanism where the majority vote among the candidate solutions determines the final solution:

$$\hat{C}^{\text{final}}_{\text{new}} = \text{mode}\left(\{\hat{C}^{(i)}_{\text{new}}\}_{i=1}^{N}\right)$$

In the context of automated optimization problem-solving, the ensemble strategy can be particularly effective when combined with other prompting techniques discussed earlier, such as few-shot prompting, self-consistency prompting, chain-of-thought prompting, and tree-of-thought prompting. By integrating the outputs from multiple models employing these techniques, we can achieve better performance and reliability, further advancing the capabilities of large language models in solving complex problems.

### 3.3. Repairing Incorrect Output

In many cases, the initial output generated by the model may contain errors or suboptimal solutions. To address this, we employ two symbolic approaches using `SymPy` (Meurer et al., 2017) to repair and refine the generated solutions, ensuring better accuracy.

**Algorithm 2** Automated Condition Repair
| |
| --- |
| 1: **Input:** Constraint inequality $E$ |
| 2: **Output:** Transformed inequality without division |
| 3: **if** $E$ does not have any division **then** |
| 4:   **return** $E$ |
| 5: **end if** |
| 6: $L := E$.lhs |
| 7: $R := E$.rhs |
| 8: $O := E$.relation |
| 9: $D := 1$ |
| 10: **for** each term $T$ in $L$ and $R$ **do** |
| 11:   **if** $T$ has a denominator **then** |
| 12:     $d := T$.denominator |
| 13:     $D := D \times d$ |
| 14:   **end if** |
| 15: **end for** |
| 16: $L := \text{Simplify}(L \times D)$ |
| 17: $R := \text{Simplify}(R \times D)$ |
| 18: **return** $\text{Inequality}(L, O, R)$ |

```
{
  "id": <unique integer>,
  "question": "<Problem Statement>",
  "code": "<source code>"
  "results": {
    "<Output 1>": "<empty or number>",
    "<Output 2>": "<empty or number>",
    "<Output 3>": "<empty or number>",
    "<Output 4>": "<empty or number>",
    ...
  }
},
```

*Figure 1.* The data format

**Fixing Incorrect Conditions**   In optimization problems, modeling constraints with mathematical inequalities is crucial. However, many Linear Programming Solvers only support weak inequalities (i.e., $\leq$ or $\geq$). This leads to issues when the problem statements include terms such as "*less than*" or "*more than*", causing the model to produce strict inequalities (i.e., $<$ or $>$), resulting in `Runtime Error` exception. To mitigate this problem, we symbolically transform strict inequalities of two types $E_1 > E_2$ and $E_1 < E_2$ into the weak version $E_1 \geq E_2 + 1$ and $E_1 \leq E_2 - 1$, assuming that both $E_1$ and $E_2$ are all integers.

**Fixing Incorrect Division Expressions**   Optimization problems often require modeling ratios involving decision variables, making constraints or objective functions non-linear. Non-linear expressions can pose challenges, as many optimization solvers, particularly linear programming solvers, cannot handle them directly. To correct the issue, we symbolically remove all division operators involving decision variables by deriving equivalent transformations.

- Fixing objective function: If it contains division between a decision variable and a constant, i.e., $\frac{v}{c}$, where $v$ is the variable and $c$ is the constant, we transform it into $\frac{1}{c}v$.

- Fixing constraint inequality: If an inequality contains a division operator, we transform the inequality using Algorithm 2 to remove all divisions. The algorithm extracts two sides of the inequalities, finds the common multiple $D$, and multiplies $D$ on both sides. This approach removes all divisions on both sides of the inequality while keeping it equivalent to the original one.

# 4. Experiment

## 4.1. Data Set

To evaluate our approaches, we use the newly published data in the Automated Optimization Problem-solving With Code Challenge at ICML 2024 AI4Math Workshop (Huang et al.). The data set consists of two partitions: a Train set and a Test set. The training set and the testing set contain 1025 and 421 Linear Programming problems respectively with various difficulties. Some of the data points in the training set were synthesized by GPT-3.5 and GPT-4, which would introduce some degree of noise to the training process. However, the organizer assures that all data points in the testing set are valid.

The training data is stored in `JSON` format with the fields: `id`, `question`, `code`, and `results` representing the id, optimization problem statement, the sample solution, and the required outputs. The testing set is also stored in `JSON` format with the fields: `id`, `question`, `results` represent the id, optimization problem statement, and the required outputs respectively. The data format can be found in Figure 1. The task was to fill in the required blanks in the `results` field of the testing data, and the model's performance was evaluated with answer accuracy.

## 4.2. Implementation Details

To create few-shot examples, we sample the optimization problems and their corresponding solution randomly and uniformly from the training data set. To create intermediate reasoning steps, we instruct `Llama-3-8B-Instruct` to produce reasoning steps before generating the final Python code. The final intermediate reasoning steps were manually analyzed and selected after some iterations. All of the experiments were implemented in Python using the VLLM framework (Kwon et al., 2023). The models' inferences were able to run on a single RTX 3090 GPU, apart from

| TECHNIQUE | ACCURACY |
|---|---|
| LLaMA-3-8B-Instruct + 5-SHOT | 57.48% |
| LLaMA-3-8B-Instruct + SELF-CONSISTENCY (5-SHOT, $k = 15$) | 60.81% |
| LLaMA-3-8B-Instruct + CHAIN OF THOUGHTS (5-SHOT, $k = 15$) | **78.15**% |
| LLaMA-3-8B-Instruct + TREE OF THOUGHTS (5-SHOT, $b_1 = 3, b_2 = 5$) | 62.00% |
| DeepSeekCoder-7B-Instruct + CHAIN OF THOUGHTS (2-SHOT, $k = 15$) | 71.50% |
| DeepSeekCoder-7B-Instruct + TREE OF THOUGHTS (2-SHOT, $b_1 = 3, b_2 = 5$) | 61.52% |
| DeepSeekCoder-33B-Instruct + TREE OF THOUGHTS (5-SHOT, $b_1 = 1, b_2 = 5$) | 62.95% |
| ENSEMBLE STRATEGY | **80.52**% |

*Table 1.* Performance of different prompting techniques with open-source LLMs.

DeepSeekCoder-33B-Instruct, which we need 8 GPUs to run inference on. The full implementation can be found at https://github.com/kurone02/AutoLP.

### 4.3. Experimental Results

We have conducted experiments demonstrating the performance of different prompting techniques using open-source large language models. The accuracy of each technique is presented in Table 4.1. The baseline performance using 5-shot prompting with LLaMA-3-8B-Instruct achieves an accuracy of $57.48\%$, showcasing that even with minimal context, a generic LLM like LLaMA-3 is able to perform complex mathematical reasoning. Introducing the self-consistency mechanism shows a modest increase in accuracy to $60.81\%$. Incorporating a Chain of Thoughts with self-consistency significantly boosts the model's accuracy by nearly $18\%$, proving the importance of intermediate reasoning steps. Surprisingly, the Tree of Thoughts method cannot outperform COT, falling short by a wide margin. Our hypothesis for the underperformance of TOT is that the current state of Large Language Models could not self-verify such complex reasoning path (Kambhampati et al., 2024), thus, the evaluator $\mathcal{E}$ is not a reliable heuristic.

We also experimented with another LLM that is specifically fine-tuned for code-related tasks, the DeepSeekCoder series (Guo et al., 2024). Due to a smaller context size, the 7B parameters variant can only process 2 examples. Nevertheless, the model achieves good performance at $71.5\%$ with the COT technique and comparable results to LLaMA-3 in using TOT. The performance in TOT can be explained by better mathematical ability from DeepSeekCoder, making it comparable to LLaMA-3 even when provided with fewer examples. We also conducted experiment with DeepSeekCoder-33B-Instruct, however, due to limited resources, we cannot fully test other prompting techniques other than TOT, which boosts the performance of TOT by a small margin.

Finally, we assemble the results obtained from various methods and models to produce the final answers, achieving $81\%$ accuracy, ranked $10^{th}$ place on the private leaderboard.

## 5. Future Work

Even though open-source LLMs demonstrate good experimental results in the challenge, there are important areas needing further investigation.

- **Improvement of Evaluation Mechanisms**. The underperformance of ToT prompting suggests a need for better evaluation mechanisms within LLMs. Future work can focus on improving the self-evaluation capability of LLMs by training with synthetic data similar to Chen et al. (2024) or utilizing RAG-based mechanism (Wei et al., 2022b).

- **Fine-tuning on the training data set**. The performance can be further enhanced by fine-tuning an LLM or a small model on the competition training data set, which we have omitted in this technical report due to the lack of resources.

- **Incorporate more difficult questions in the data set**. Most of the questions in the data set are Linear Programming problems, lacking diversity in difficulties. Future work could focus on building a more diverse data set consisting of other types of problems in optimization such as Convex Optimization, Dynamic Programming, or Stochastic Optimal Control.

## 6. Conclusion

This technical report has investigated the capabilities of open-source Large Language Models (LLMs) in formulating and solving optimization problems through various prompting techniques. The exploration of methods such as few-shot prompting, self-consistency prompting, chain-of-thought (CoT) prompting, and tree-of-thought (ToT) prompting has provided valuable insights into how LLMs can be effectively utilized for complex mathematical reasoning tasks.

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

# Appendix

## A. Prompt templates

### A.1. Few-shot Chain-of-Thought Prompts

In this section, we provide the prompts used in our technical report. Only the Chain-of-Thought template will be provided as the Few-shot version can be easily obtained by removing the reasoning text from the prompt.

### System Prompt

You are an Operational Research Professor who can solve optimization problems with python code.

### Example 1

**User:** Give a step by step reasoning and python code to solve the following Optimization problem. Wrap the print statements between "## start solving" and "## end solving". Only print what the requirements requires.

A company is organizing a team-building event and needs to assign participants to different activities. They have a total of 100 employees. Activity A requires 5 employees a team, activity B requires 3 employees a team, and activity C requires 7 employees a team. The company has a total of 100 employees available for the event. The company has a limitation on the number of teams in activity B, which cannot exceed 20. The company wants to maximize participation teams and decides to allocate different weights to each activity: activity A has a weight of 3, activity B has a weight of 2, and activity C has a weight of 4. The objective is to maximize the total participation weighted by the assigned weights.

Requirements:

- The number of teams in activity A: ?

- The number of teams in activity B: ?

- The number of teams in activity C: ?

- The total weighted participation: ?

**Assistant:** Let's think step by step.

We want to maximize the total participation weighted by the assigned weights. So, we can set up a linear programming problem.

Let's define the variables:

* NumParticipantsA: The number of participants in activities A

* NumParticipantsB: The number of participants in activities B

* NumParticipantsC: The number of participants in activities C

Note that "The objective is to maximize the total participation weighted by the assigned weights". Therefore, we need to MAXIMIZE the objective function.

Since activity A has a weight of 3, activity B has a weight of 2, and activity C has a weight of 4, the objective function is:
`3 * num_teams_A + 2 * num_teams_B + 4 * num_teams_C`

The objective function is subjected to the following constraints:

1. Total participants cannot exceed 100, and Activity A requires 5 employees a team, activity B requires 3 employees a team, and activity C requires 7 employees a team.

`5 * num_teams_A + 3 * num_teams_B + 7 * num_teams_C <= 100`

2. `num_teams_A >= 0`

3. `num_teams_B >= 0`

4. `num_teams_C >= 0`

5. The company has a limitation on the number of teams in activity B, which cannot exceed 20. `num_teams_B <= 20`

Now, we can start solving the problem with python using pulp library

```python
# Import PuLP library
from pulp import *

# Define the decision variables
# number of teams in activity A
num_teams_A = LpVariable("NumTeamsA", lowBound=0, cat='Integer')
# number of teams in activity B
num_teams_B = LpVariable("NumTeamsB", lowBound=0, upBound=20, cat='Integer')
# number of teams in activity C
num_teams_C = LpVariable("NumTeamsC", lowBound=0, cat='Integer')

# Define the question as a maximum or minimum problem
problem = LpProblem("TeamBuildingEvent", LpMaximize)

# Define the objective function
objective = 3 * num_teams_A + 2 * num_teams_B + 4 * num_teams_C
# maximize the total participation weighted by the assigned weights
problem += objective

# Define the constraints
# the total number of employees is 100
problem += 5 * num_teams_A + 3 * num_teams_B + 7 * num_teams_C <= 100

# Solve the problem
status = problem.solve()

# Output the answer
print("## start solving")
print("The number of teams in activity A:", num_teams_A.value())
print("The number of teams in activity B:", num_teams_B.value())
print("The number of teams in activity C:", num_teams_C.value())
print("The total weighted participation:", objective.value())
print("## end solving")
```

## Example 2

**User:** Give a step by step reasoning and python code to solve the following Optimization problem. Wrap the print statements between "## start solving" and "## end solving". Only print what the requirements requires.

BrightBulb LLC manufactures LED and incandescent bulbs. They have 3 different machines for production. Each machine has a different production capacity and cost associated with running it per day. The following Table provides details on the capacity (bulbs per day) and cost (cost per day) for each machine.

Table (Capacity and cost for each machine):

| Bulb Type | Machine 1 | Machine 2 | Machine 3 |
| --------- | --------- | --------- | --------- |
| LED (Capacity) | 300 | – | 500 |
| Incandescent (Capacity) | – | 400 | – |
| Cost per Day | $200 | $250 | $350 |

BrightBulb LLC wants to minimize the total cost of producing 5000 LED bulbs and 6000 incandescent bulbs per week.

Machine 1 and Machine 3 are limited to LED bulb production and Machine 2 is limited to incandescent bulb production.

The weekly production target for LED bulbs is at least 5000, and for incandescent bulbs is at least 6000.

Each machine can only operate for a maximum of 5 days a week due to maintenance requirements.

Requirements:

- The number of LED bulbs produced on machine 1: ?

- The number of incandescent bulbs produced on machine 2: ?

- The number of LED bulbs produced on machine 3: ?

- The total cost of production: ?

**Assistant:** Let's think step by step.

This problem can be formulated as a linear programming problem to minimize the total cost of producing 5000 LED bulbs and 6000 incandescent bulbs per week.

Let's define the variables:

* `num_led_on_machine1`: The number of LED bulbs produced on machine 1 per day * `num_led_on_machine3`: The number of LED bulbs produced on machine 3 per day * `num_inc_on_machine2`: The number of incandescent bulbs produced on machine 2 per day

To derive the objective function, note that - Machine 1 has capacity of 300 with $200 cost - Machine 3 has capacity of 500 with $350 cost - Machine 2 has capacity of 400 with $250 cost

Therefore, the objective function is to minimize the total cost:

Minimize:

```
objective = 200 / 300 * num_led_on_machine1 + 250 / 400 * num_inc_on_machine2
+ 350 / 500 * num_led_on_machine3|
```

Subject to the following constraints:

1. The total number of LED bulbs produced must be at least 5000:

```
num_led_on_machine1 + num_led_on_machine3 >= 5000
```

2. The total number of incandescent bulbs produced must be at least 6000: `num_inc_on_machine2 >= 6000`

3. Machine 1 and Machine 3 are limited to LED bulb production:

`num_led_on_machine1 + num_led_on_machine3 <= 300 * 5` (machine 1 can produce up to 300 bulbs per day, and it's limited to 5 days a week)

`num_led_on_machine1 + num_led_on_machine3 <= 500 * 5` (machine 3 can produce up to 500 bulbs per day, and it's limited to 5 days a week)

4. Machine 2 is limited to incandescent bulb production: `num_inc_on_machine2 <= 400 * 5` (machine 2 can produce up to 400 bulbs per day, and it's limited to 5 days a week)

5. Each machine can only operate for a maximum of 5 days a week:

```
num_led_on_machine1 <= 5
```

```
num_led_on_machine3 <= 5
```

```
num_inc_on_machine2 <= 5
```

Now, we can start solving the problem with python using pulp library

```
1  from pulp import *
2
3  # Define the variables
4  num_led_on_machine1 = LpVariable("NumLEDOnMachine1", lowBound=0, cat='Integer')
5  num_led_on_machine3 = LpVariable("NumLEDOnMachine3", lowBound=0, cat='Integer')
6  num_inc_on_machine2 = LpVariable("NumINCOnMachine2", lowBound=0, cat='Integer')
```

```
7
8  # Define the objective function
9  problem = LpProblem("BrightBulbProblem", LpMinimize)
10 objective = 200 / 300 * num_led_on_machine1 + 250 / 400 * num_inc_on_machine2 + 350 / 500
       * num_led_on_machine3
11 problem += objective
12
13 # Define the constraints
14 problem += num_led_on_machine1 + num_led_on_machine3 >= 5000
15 problem += num_inc_on_machine2 >= 6000
16 problem += num_led_on_machine1 + num_led_on_machine3 <= 300 * 5
17 problem += num_led_on_machine1 + num_led_on_machine3 <= 500 * 5
18 problem += num_inc_on_machine2 <= 400 * 5
19 problem += num_led_on_machine1 <= 5
20 problem += num_led_on_machine3 <= 5
21 problem += num_inc_on_machine2 <= 5
22
23 # Solve the problem
24 status = problem.solve()
25
26 # Output the answer
27 print("## start solving")
28 print("The number of LED bulbs produced on machine 1:", value(num_led_on_machine1))
29 print("The number of incandescent bulbs produced on machine 2:", value(num_inc_on_machine2
       ))
30 print("The number of LED bulbs produced on machine 3:", value(num_led_on_machine3))
31 print("The total cost of production:", value(objective))
32 print("## end solving")
```

## Example 3

**User:** Give a step by step reasoning and python code to solve the following Optimization problem. Wrap the print statements between "## start solving" and "## end solving". Only print what the requirements requires.

IND Foods factory produces basmati rice and bananas. To make one kilogram of basmati rice requires 1.5 hours of human labor, 2 hours of machine work, and 3 hours of resting under the sun. To make one kilogram of bananas requires 2 hours of human labor, 4 hours of machine work, and 1.5 hours of resting under the sun. The factory only has 90 hours of human labor, 150 hours of machine labor, and 70 hours of time under the sun available. The net profit per kilogram of basmati rice is $15 and the net profit per kilogram of bananas is $28. How many kilograms of each should the factory make in order to maximize profit?

Requirements:

- The number of kilograms of basmati rice to produce: ?

- The number of kilograms of bananas to produce: ?

- The maximum profit: ?

**Assistant:** Let's think step by step.

Define the variables:

* `num_rice`: The number of kilograms of basmati rice to produce

* `num_bananas`: The number of kilograms of bananas to produce

Note that they can be a real number, i.e., of Continuous type.

We want to maximize the total profit, which is the sum of the profit from producing basmati rice and bananas:

Maximize: `15 * num_rice + 28 * num_bananas`

Subject to the following constraints:

1. Human labor: `1.5 * num_rice + 2 * num_bananas <= 90` (hours of human labor available)

2. Machine labor: `2 * num_rice + 4 * num_bananas <= 150` (hours of machine labor available)

3. Resting under the sun: `3 * num_rice + 1.5 * num_bananas <= 70` (hours of time under the sun available)

We can use the PuLP library to solve this problem:

```python
# Import PuLP library
from pulp import *

# Define the variables
num_rice = LpVariable("NumRice", lowBound=0, cat='Continuous')
num_bananas = LpVariable("NumBananas", lowBound=0, cat='Continuous')

# Define the problem
problem = LpProblem("INDFoodsProblem", LpMaximize)
objective = 15 * num_rice + 28 * num_bananas
problem += objective

# Define the constraints
problem += 1.5 * num_rice + 2 * num_bananas <= 90
problem += 2 * num_rice + 4 * num_bananas <= 150
problem += 3 * num_rice + 1.5 * num_bananas <= 70

# Solve the problem
status = problem.solve()

# Output the answer
print("## start solving")
print("The number of kilograms of basmati rice to produce:", value(num_rice))
print("The number of kilograms of bananas to produce:", value(num_bananas))
print("The maximum profit:", value(objective))
print("## end solving")
```

## Example 4

**User:** Give a step by step reasoning and python code to solve the following Optimization problem. Wrap the print statements between "## start solving" and "## end solving". Only print what the requirements requires.

A food truck wants to make bean burritos and beef burritos using 5000 grams of lettuce. The profit per bean burrito is $6.5 and the profit per beef burrito is $9. The bean burrito contains 25 grams of lettuce and the beef burrito contains 18 grams of lettuce. The beef burrito is much more popular and at least four times the amount of beef burritos need to be made than the bean burritos. However, a minimum of 5 bean burritos need to be made. How many of each burrito should be made in order to maximize profit?

Requirements:

- The number of bean burritos: ?

- The number of beef burritos?: ?

- Total profit: ?

**Assistant:** Let's think step by step.

Define the variables:

* `num_bean_burritos`: The number of bean burritos of Integer type

* `num_beef_burritos`: The number of beef burritos of Integer type

We want to maximize the total profit, which is the sum of the profit from bean and beef burritos:

Maximize: `6 * num_bean_burritos + 9 * num_beef_burritos`

Subject to the following constraints:

1. Using 5000 grams of lettuce: `25 * num_bean_burritos + 18 * num_beef_burritos <= 5000`

2. Four times the amount of beef burritos need to be made than the bean burritos:

`num_beef_burritos >= 4 * num_bean_burritos`

3. Minimum of 5 bean burritos need to be made: `num_bean_burritos >= 5`

We can use the PuLP library to solve this problem:

```python
# Import PuLP library
from pulp import *

# Define the decision variables
# number of bean burritos
num_bean_burritos = LpVariable("NumBeanBurritos", lowBound=5, cat='Integer')
# number of beef burritos
num_beef_burritos = LpVariable("NumBeefBurritos", lowBound=0, cat='Integer')

# Define the question as a maximum or minimum problem
problem = LpProblem("FoodTruckProfitMaximization", LpMaximize)

# Define the objective function
# maximize the total profit from selling burritos
objective = 6.5 * num_bean_burritos + 9 * num_beef_burritos
problem += objective

# Define the constraints
# lettuce constraint
problem += 25 * num_bean_burritos + 18 * num_beef_burritos <= 5000
# beef burrito popularity constraint
problem += num_beef_burritos >= 4 * num_bean_burritos
# minimum bean burrito constraint
problem += num_bean_burritos >= 5

# Solve the problem
status = problem.solve()

# Output the answer
print("## start solving")
print("The number of bean burritos:", num_bean_burritos.value())
print("The number of beef burritos:", num_beef_burritos.value())
print("Total profit:", objective.value())
print("## end solving")
```

### Example 5

**User:** Give a step by step reasoning and python code to solve the following Optimization problem. Wrap the print statements between "## start solving" and "## end solving". Only print what the requirements requires.

A lab has 20000 mg of mRNA anti-viral available to make children's and adult vaccines. Each children's vaccine contains 50 mg of mRNA and 50 mg of fever suppressant. Each adult vaccine contains 75 mg of mRNA and 75 mg of fever suppressant. Since adult vaccines are more essential, at least 70% of vaccines should be adult vaccines. However, at least 50 children's vaccines should be made. How many of each vaccine should be made to minimize the amount of fever suppressant used?

Requirements:

- The number of children's vaccine: ?

- The number of adult vaccine: ?

- The amount of fever suppressant used: ?

**Assistant:** Let's think step by step.

Define the variables:

* `num_children_vaccines`: The number of children's vaccines * `num_adult_vaccines`: The number of adult vaccines

We want to minimize the amount of fever suppressant used, which is the sum of the fever suppressant used in children's and adult vaccines:

Minimize: `50 * num_children_vaccines + 75 * num_adult_vaccines`

Subject to the following constraints:

1. Total amount of mRNA anti-viral available:

`50 * num_children_vaccines + 75 * num_adult_vaccines <= 20000`

2. At least 70% of vaccines should be adult vaccines:

`num_adult_vaccines >= 0.7 * (num_children_vaccines + num_adult_vaccines)`

3. At least 50 children's vaccines should be made: `num_children_vaccines >= 50`

Now, we can start solving the problem with python

```python
# Import PuLP library
from pulp import *

# Define the decision variables
# number of children's vaccines
num_children_vaccines = LpVariable("NumChildrenVaccines", lowBound=50, cat='Integer')
# number of adult vaccines
num_adult_vaccines = LpVariable("NumAdultVaccines", lowBound=0, cat='Integer')

# Define the question as a minimum or maximum problem
problem = LpProblem("LabVaccineProblem", LpMinimize)

# Define the objective function
# minimize the total amount of fever suppressant used
objective = 50 * num_children_vaccines + 75 * num_adult_vaccines
problem += objective

# Define the constraints
# mRNA anti-viral constraint
problem += 50 * num_children_vaccines + 75 * num_adult_vaccines <= 20000
# adult vaccine popularity constraint
problem += num_adult_vaccines >= 0.7 * (num_children_vaccines + num_adult_vaccines)
# children's vaccine constraint
problem += num_children_vaccines >= 50

# Solve the problem
status = problem.solve()

# Output the answer
print("## start solving")
print("The number of children's vaccines:", num_children_vaccines.value())
print("The number of adult vaccines:", num_adult_vaccines.value())
print("The amount of fever suppressant used:", objective.value())
print("## end solving")
```

### A.2. Tree-of-thought Evaluator Prompt

In this section, we provide the prompt used for the neural evaluator $\mathcal{E}$ utilizing LLMs to grade its own solutions. We limit the output when evaluating to only 1 token, and manually increase the logit of the tokens corresponding to the number 0 to 9 by an arbitrary big number i.e., 10000, to ensure the output is always valid.

**System Prompt**

You are an Operational Research Professor who needs to grade students' solution.

**Evaluator Prompt**

You will be given a Linear Programming problem and the current solution steps that might not completed yet. Give the solution a score from 0 (bad) to 9 (good) that represents how good the reasoning is to solve the problem.

The student's solution will contains two part: reasoning and python code.

In both reasoning and python code, there are 4 main sections:

- Define the decision variables

- Define the question as a maximum or minimum problem

- Define the objective function

- Define the constraints

####Problem####

{problem}

####Current Solution####

{solution}

