# OpenReview forum: "Technical Report for ICML 2024 Automated Math Reasoning Challenge: Solving Optimization Problems with Open Source Large Language Model"
_ICML.cc/2024/Workshop/AI4MATH — ICML 2024 Workshop AI4MATH Poster_

### Official Review · Reviewer_RM5c · 2024-06-09

**Rating:** 4
**Confidence:** 4

**Summary:**

The paper explores the capabilities of open-source Large Language Models (LLMs) to solve optimization problems using different prompting techniques. It contributes to the field of automated mathematical reasoning, specifically focusing on optimization problems.

**Questions:**

1. Can you discuss some of the limitations or failure cases encountered during the experimental phase that did not perform as expected?
2. Since the methods used are all existing, how do you envision this work impacting the field of automated mathematical reasoning?

**Reasons To Accept:**

1. The paper is well-written and organized. The paper has clear methodologies and results, demonstrating the potential of LLMs in mathematical reasoning.

**Reasons To Reject:**

1. The paper is mainly about prompt engineering with existing methods. There's no innovation.
2. The paper could delve deeper into the mechanisms behind why certain techniques outperform others, which could provide valuable insights into how to optimize prompt engineering for future research.
3. The paper could benefit from more analysis of the results, particularly regarding the failure cases of certain techniques.

---

### Official Review · Reviewer_bB5U · 2024-06-11

**Rating:** 7
**Confidence:** 4

**Summary:**

The paper shows how to leverage open LLMs and well studies techniques, such as few-shots or CoT, to translate the problem from natural language to the full solution.

**Questions:**

Q1. Doing everything in one shot is probably hard. It would be extremely interesting to see if it's easier to use few-shots / CoT to translate the problem from words to formal language and use again few-shots / CoT to translate from formal language to the final solution. With this breakdown, authors can determine if the bottleneck is in the Python translation or the optimization problem formulation.

Q2. It's not super clear why ToT doesn't improve compared to CoT. It would be great to see a depth-in analysis, or at least a bunch of examples where CoT works/doesn't work and ToT doesn't work/works, respectively.

Q3. In section 3.2 you always denote the final guess as mode(...). For easy and solved problems, this is fine and probably encouraged. It would be interesting to see if you consider all possible paths and validate the solution when one path (for example one of the multiple CoT) is correct.

Q4. Is there a particular reason why you use the "Let's think step by step" prompt?

**Reasons To Accept:**

S1. The section 3.2 is pretty strong, in particular on the CoT and the ensemble strategy.

S2. The section 3.3 is very well motivated, with a deep analysis on how to deal with inequalities and divisions.

S3. Overall the report is very clear and all missing code have been replaced with algorithmic explanations (i.e. Alg 1).

**Reasons To Reject:**

W1. Although you mentioned a lack of resources, it would be great to see smaller models finetuning. It's possible to finetune even with 1 GPU for less than 10 hours (i.e. 1.5B models after a LoRA finetuning).

W2. The definition of the neural evaluator $\varepsilon$ is not clear. Did you pretrain a model? Did you use a autoeval?

---

### Official Review · Reviewer_T2hc · 2024-06-11

**Rating:** 7
**Confidence:** 3

**Summary:**

The authors use two pre-trained large language models (LLaMA-3-8B-Instruct and DeepSeekCoder-7B-Instruct) to find solutions to linear programs by leveraging several prompt augmentation techniques (few-shot, self-consistency, chain-of-thought and tree-of-thought) and comparing their accuracy.

**Questions:**

n/a (addressed above)

**Reasons To Accept:**

Overall, the paper is clear and well-written. The authors give a qualitative review of past work and clearly define and explain the different prompting methods considered, as well as the changes they did and experiments they ran.

**Reasons To Reject:**

Overall, I believe a few (minor) changes and corrections would improve the clarity and readability of the manuscript:
 - 1
	 - The authors could outline their contribution more clearly at the end of the introduction (namely, listing the prompt engineering frameworks that are considered and mentioning that it is tried on LPs).
- 2.1
	- Calude -> Claude
	- "The model approached GPT-3’s performance but ten times smaller and faster" is missing a "was"
- 3.1
	- Definition 3.4: $u_s$ -> $n_u$ in $U$
	- Definition 3.4: It seems that $U_1$, $A_1$ etc correspond to different ordered sets (and not indexing into the sets), while $\mathcal{P}_i$ appears to be an indexing into $\mathcal{P}$? If so, having the same notation is a bit confusing
	- If space allows, a concise and concrete example of the different definitions would help with understanding. Since the authors test the methods on LPs, a generic LP could be a possible example.
- 3.2
	- An example of "System instruction" would be appreciated
	- A quick explanation for what "mode" / "majority voting mechanism" are would be appreciated. Is this conducted through another neural network model?
	- "we construct the tree-of-thought prompt PToT is similar to Equation 2": is similarly -> similarly
	- Eq 3: last $P_k$ -> $P_\text{new}$
	- Ensemble Strategy - "It has been known" should be "shown"?
- 3.3
	- In "Fixing Incorrect Conditions", $E2$ -> $E_2$
	- "we transform the inequality using Algorithm 2 to remove all divisions" -> The authors could add a one-sentence description of what Algorithm 2 does.
	- Are these two repairing approaches (including the algorithm) common in literature? If so, a citation would be appreciated. If this is the authors' contribution, this could be emphasized in the introduction.
- 4.1
	- "with the fields ... represent ..." -> representing
	- The task was to fill in the required blanks in the results field of the testing data -> Could the authors add an example sample LP from the dataset, to see what the "blanks" look like and what an expected Python solution looks like?
	- I am unclear on whether the LLMs try to find the solution set of the LP, or something else? Is the solution always unique here?
- 4.2
	- "we needs"
- 4.3
	- A brief discussion of how the authors' results compare to what they have observed in the literature on similar tasks would be appreciated, either in the results section or in the introduction.

---

### Official Review · Reviewer_1pWD · 2024-06-13

**Rating:** 7
**Confidence:** 4

**Summary:**

The paper investigates the ability of open-source LLMs such as Llama3 and DeepSeekCoder to solve easy optimization problems from automatically generating Python codes using few-shot learning. The paper aims at comparing different prompting techniques such as few-shot, self-consistency, chain-of-thought, and tree-of-thought in few-shot learning. The paper also proposes an evaluation module using LLMs to evaluate the quality of a reasoning path. However, the evaluation performance is not as good as expected. The results show that chain-of-thought achieves the best performance among the proposed prompting techniques.

**Questions:**

1. Please list the specific prompts used in chain-of-thought prompting method.
2. Do you also use definition 3.6 to give a score of different models in the ensemble method?

**Reasons To Accept:**

1. The paper is well-organized and well-written. It studies the performance of almost all the common prompting techniques of LLMs for solving simple optimization problems.

**Reasons To Reject:**

1. The paper doesn't show the content/template of each prompt used in different prompting techniques.

---

### Meta-Review · Area_Chair_pvpg · 2024-06-13

**Recommendation:** Accept (Poster)
**Confidence:** 3

**Metareview:**

**Paper Summary:**

The paper explores the use of two pre-trained large language models (LLaMA-3-8B-Instruct and DeepSeekCoder-7B-Instruct) to solve linear programs by leveraging several advanced prompt augmentation techniques, including few-shot, self-consistency, chain-of-thought (CoT), and tree-of-thought (ToT). The authors provide a comprehensive comparison of these techniques in terms of their accuracy and performance.



**Review Summary:**

The paper presents a valuable contribution to the field of automated mathematical reasoning by demonstrating the application of LLMs and advanced prompt engineering techniques to solve linear programs. Despite the need for minor revisions and deeper analysis in certain areas, the paper's strengths in clarity, organization, and methodological rigor make it a worthy addition to the literature. With the suggested improvements, the paper could significantly impact the understanding and application of prompt engineering in mathematical problem-solving.

**Reasons to Accept:**

1. The paper is clear, well-written, and well-organized, making it accessible and easy to follow. The methodologies and results are presented in a structured manner, contributing to the overall readability of the manuscript.
2. The authors provide a thorough qualitative review of past work, clearly defining and explaining the different prompting methods considered. They also detail the modifications and experiments conducted, enhancing the reader's understanding of the study.
3. Sections 3.2 and 3.3 are particularly strong. Section 3.2 provides an in-depth analysis of CoT and the ensemble strategy, while Section 3.3 is well-motivated with a detailed examination of how to handle inequalities and divisions.
4. Missing code is effectively replaced with algorithmic explanations, such as in Algorithm 1, providing clarity and maintaining the technical rigor of the paper.

**Reasons to Reject:**

1.  A few minor changes and corrections could enhance the manuscript's clarity and readability. These include outlining the authors' contributions more explicitly in the introduction, correcting typographical errors, and providing more concrete examples and explanations for certain concepts.
2. The definition and implementation of the neural evaluator ($\varepsilon$) are not entirely clear. Further details on whether a pretrained model or auto-evaluation was used would be beneficial.
3. The paper lacks depth in explaining why certain prompting techniques outperform others. A deeper analysis of the mechanisms and failure cases would provide valuable insights and improve the overall contribution to the field.

---

### Decision · Program_Chairs · 2024-06-13

Accept (Poster)